# Unsupervised multi-source domain adaptation with no observable source data

**Hyunsik Jeon, Seongmin Lee, U Kang** *

Seoul National University, Seoul, Republic of Korea

\* ukang@snu.ac.kr

## Abstract

Given trained models from multiple source domains, how can we predict the labels of unlabeled data in a target domain? Unsupervised multi-source domain adaptation (UMDA) aims for predicting the labels of unlabeled target data by transferring the knowledge of multiple source domains. UMDA is a crucial problem in many real-world scenarios where no labeled target data are available. Previous approaches in UMDA assume that data are observable over all domains. However, source data are not easily accessible due to privacy or confidentiality issues in a lot of practical scenarios, although classifiers learned in source domains are readily available. In this work, we target data-free UMDA where source data are not observable at all, a novel problem that has not been studied before despite being very realistic and crucial. To solve data-free UMDA, we propose DEMS (Data-free Exploitation of Multiple Sources), a novel architecture that adapts target data to source domains without exploiting any source data, and estimates the target labels by exploiting pre-trained source classifiers. Extensive experiments for data-free UMDA on real-world datasets show that DEMS provides the state-of-the-art accuracy which is up to 27.5% point higher than that of the best baseline.

## Introduction

Given trained models from multiple source domains, how can we predict the labels of unlabeled data in a target domain? Unsupervised multi-source domain adaptation (UMDA) aims at predicting the labels of unlabeled target data by utilizing the knowledge of multiple source domains. Many previous works [1–9] for UMDA have focused on finding domain-invariant features $z$ of data $x$ to transfer the knowledge of conditional probability $p(y|z)$, where $y$ represents the label of data $x$, from the source domains to the target domain. It is thus essential for UMDA that data $x$ is observable in all domains to be able to estimate the conditional probabilities $p(z|x)$ of all domains while finding the domain-invariant features $z$.

However, source data are not always accessible, although models of conditional probabilities $p(y|x)$ learned in source domains are often readily available, due to privacy or confidentiality issues in many practical scenarios. For instance, a hospital is allowed to access disease classifiers that are trained in other hospitals but not the data the classifiers observed because of privacy issues. Fig 1 illustrates the UMDA problems with two different constraints. It is

**

**Data Availability Statement:** The data and code are available at: https://github.com/snudatalab/DEMS.

**Funding:** This work was supported by Institute of Information & communications Technology

Planning & Evaluation(IITP) grant funded by the Korea government(MSIT) (No.2020-0-00894, Flexible and Efficient Model Compression Method for Various Applications and Environments). The Institute of Engineering Research and ICT at Seoul National University provided research facilities for this work. The funders had no role in study design, data collection and analysis, decision to publish, or preparation of the manuscript.

**Competing interests:** The authors have declared that no competing interests exist.

problematic to find a shared manifold $z$ and to translate data between domains if source data are not observable at all (Fig 1b), compared to the setting where data are observable in all domains (Fig 1a).

In this paper, we focus on data-free UMDA (Fig 1b), a more difficult but practical problem of knowledge transfer from multiple source domains to an unlabeled target domain. The main challenges are that: 1) we cannot directly estimate the target conditional probability $p(y|x)$ since target labels are not given, and 2) we cannot directly learn the shared manifold $z$ between domains since there is no information of source domain data distributions $p(x)$. We propose DEMS (Data-free Exploitation of Multiple Sources), a novel architecture that adapts target data to source domains without using any source data and estimates the target labels exploiting pre-trained source classifiers. To the best of our knowledge, there has been no approach for data-free UMDA.

Table 1 compares DEMS with other algorithms for data-free UMDA in various perspectives. Since data-free UMDA is a new problem without previous studies, we introduce several

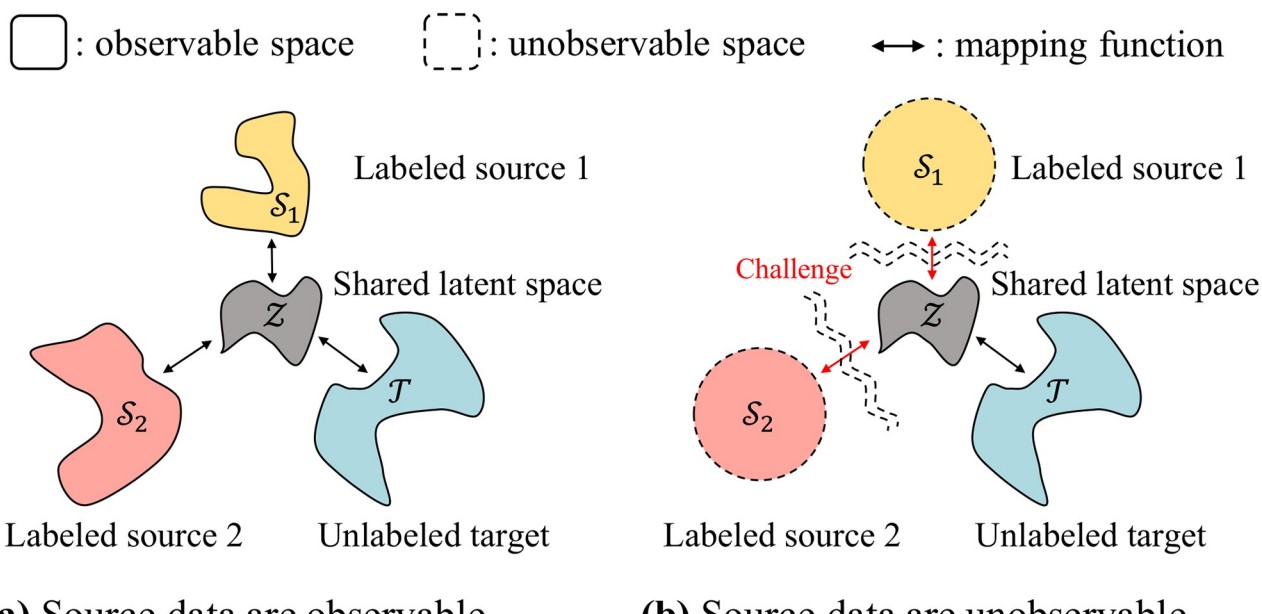

**Fig 1. An illustration of unsupervised multi-source domain adaptation (UMDA) problems.** (a) illustrates UMDA problem with observable source data, and (b) illustrates data-free UMDA problem with no observable source data. It is challenging to reduce the distribution discrepancy between source and target domains in (b) since there are no accessible source data.

**Table 1. Comparison of DEMS and other methods.**

| Method | Utilize multiple sources | Consider domain proximity | Domain adaptation |
|---|---|---|---|
| Best Single Source | X | X | X |
| Average | **O** | X | X |
| Weighted Sum | **O** | **O** | X |
| DEMS (proposed) | **O** | **O** | **O** |

DEMS is the only method supporting all the desired properties.

**Table 2. Table of frequently-used symbols.**

| Symbol | Description |
|:---:|:---|
| $\mathcal{T}$ | Target domain |
| $\mathcal{S}_k$ | $k$-th source domain |
| $M_{\mathcal{S}_k}$ | $k$-th source classifier |
| $x_{\mathcal{T}}, y_{\mathcal{T}}$ | Data and label of target domain |
| $x_{\mathcal{S}_k}, y_{\mathcal{S}_k}$ | Data and label of $k$-th source domain |
| $A_k$ | Adaptation model from target domain to $k$-th source domain |
| $E$ | Encoder |
| $D_{\mathcal{T}}$ | Decoder for target domain |
| $D_{\mathcal{S}_k}$ | Decoder for $k$-th source domain |

baselines. The first one is *Best Single Source* which employs source classifiers individually and to find the best source classifier. The second one is *Average* which averages the results of all source classifiers. The third one is *Weighted Sum* which combines the results of all source classifiers by calculating domain proximities in a heuristic way. DEMS is the only method that utilizes multiple sources, considers domain proximity, and adapts source domains into target domain. Table 2 lists the symbols used in this paper. The contributions of this work are as follows:

- **Problem Formulation.** We formulate a new problem of data-free UMDA which is challenging but important task for transfer learning (see Fig 1b). Unlike traditional UMDA, data-free UMDA needs to handle the issue of inaccessible source data.

- **Approach.** We propose DEMS, a novel approach to solve data-free UMDA. DEMS adapts target data to source domains and exploits given source classifiers based on our proposed domain proximity. DEMS learns the adaptation functions while regulating the classification results of the source classifiers after adaptation.

- **Performance.** Our extensive experiments demonstrate that DEMS provides the state-of-the-art accuracy which is up to 27.5% point higher than that of the best baseline (see Fig 2).

## Related work

Domain adaptations (DA) aim at transferring the knowledge of a source domain to a different but related target domain. Unsupervised domain adaptation (UDA) aims to leverage a labeled source domain dataset for label prediction for an unlabeled target domain dataset. Various approaches for UDA have been proposed including adversarial methods [10–13], distance-based methods [14–18], and optimal transportations [19, 20].

Recent works [1–9] address unsupervised multi-source domain adaptation (UMDA) which aims at transferring the knowledge from multiple source domains rather than a single one to an unlabeled target domain. UMDA bestows high potential of a superior performance by exploiting multiple source domain knowledge, but poses challenges of reducing domain discrepancy between multiple domains and obtaining appropriate domain-invariant features. Many previous works have tackled UMDA problems with various approaches. Table 3 summarizes the key differences in various approaches. Zhao et al. [5] propose an adversarial network based approach with generalization bounds for UMDA. Xu et al. [6] propose Deep Cocktail Network which addresses the domain and category shifts among multiple source domains in a multi-way adversarial manner. Peng et al. [9] introduce moment matching to

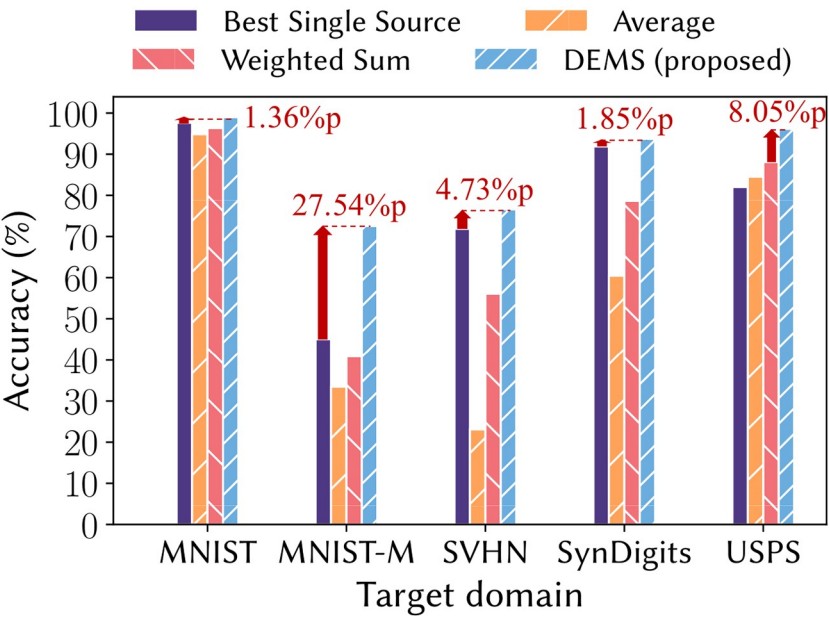

**Fig 2. Classification accuracy.** DEMS shows the best classification accuracy for five target domains; each percentage indicates the accuracy increase compared to the second-best one for each target domain.

UMDA to dynamically align moments of low-dimensional features in source and target domains while training source classifiers. However, these approaches assume that source data are observable and train adaptation networks to align manifolds of source and target domains. Thus they are not applicable to our setting where no source data are accessible due to strict privacy or confidentiality issues. On the other hand, DEMS trains adaptation networks using target data while regulating the results of the given source classifiers.

## Proposed method

### Problem definition

Suppose there are $N$ source domains $\mathcal{S}_1, \mathcal{S}_2, \ldots, \mathcal{S}_N$ and one target domain $\mathcal{T}$ where all domains have different data distributions. We are given pre-trained source classifiers $\{M_{\mathcal{S}_k} : x_{\mathcal{S}_k} \rightarrow y_{\mathcal{S}_k}\}_{k=1}^{N}$ that predict the labels of data from the corresponding source domains $\{\mathcal{S}_k\}_{k=1}^{N}$, and an unlabeled target dataset $X_{\mathcal{T}} = \{x_{\mathcal{T}}^i\}_{i=1}^{n_{\mathcal{T}}}$ from the target domain $\mathcal{T}$; for simplicity, we assume the target dataset is sampled from uniform label distribution. Each source classifier $M_{\mathcal{S}_k}$ is trained under a labeled dataset $\{(x_{\mathcal{S}_k}^i, y_{\mathcal{S}_k}^i)\}_{i=1}^{n_{\mathcal{S}_k}}$ which is drawn from the

**Table 3. Comparison of different latent space transformation methods for unsupervised multi-source domain adaptation (UMDA).**

| Method | Source data accessibility | Feature alignment method |
|---|---|---|
| [5, 6] | Accessible | Adversarial approach |
| [9] | Accessible | Discrepancy-based approach |
| DEMS (proposed) | Inaccessible | Source classifier-based approach |

Previous studies have proposed adversarial and discrepancy-based approaches which necessitate source data. On the other hand, DEMS works without source data by carefully utilizing source classifiers.

corresponding domain data distribution $p_{\mathcal{S}_k}(x, y)$. Note that the source datasets are unavailable to us, and only the source classifiers are available. In this work, we assume 1) *homogeneity* which indicates that sources and target domains have similar feature spaces and label distributions, and 2) *closed label set*, i.e. $y_{\mathcal{S}_k}, y_{\mathcal{T}} \in \mathcal{Y}$ for $k = 1, 2, \ldots, N$, where $\mathcal{Y}$ is the label space, indicating all domains have the same label space. The goal of *data-free UMDA* is to accurately predict the target domain labels $Y_{\mathcal{T}} = \{y_{\mathcal{T}}^i\}_{i=1}^{n_{\mathcal{T}}}$ of the corresponding target domain data $X_{\mathcal{T}} = \{x_{\mathcal{T}}^i\}_{i=1}^{n_{\mathcal{T}}}$.

## Method overview

In UMDA, directly training a target classifier $M_{\mathcal{T}} : x_{\mathcal{T}} \rightarrow y_{\mathcal{T}}$ from the target dataset is not possible since the target labels are not observable. Thus, most UMDA methods train $N$ adaptation functions $\{A_k : X_{\mathcal{T}} \rightarrow X_{\mathcal{S}_k}\}_{k=1}^N$ and exploit the pre-trained source classifiers $\{M_{\mathcal{S}_k}\}_{k=1}^N$ to predict the target labels $Y_{\mathcal{T}}$ of the target data $X_{\mathcal{T}}$. However, in data-free UMDA, we face the challenge of defining the objective function to train the adaptation functions $\{A_k\}_{k=1}^N$, sincethe source data are unobservable and we have no information about the source data distribution $p_{\mathcal{S}_k}(x)$ that was used to train $M_{\mathcal{S}_k}$.

To address the challenge, we propose DEMS (Data-free Exploitation of Multiple Sources), a novel method for unsupervised multiple domain adaptation problem when the source data are entirely unavailable. We cannot directly learn the adaptation results of the target data to the source domains since we have no information on the source domains at all. Hence, we regulate the classification results using the source classifiers instead of learning the translation between the target and the source domains directly.

We introduce four ideas in DEMS to regulate the classification results.

- The first idea is *label consistency regularization* which regulates the label predictions of all source classifiers to be similar. The adapted examples from the target domain to the source domains should all have the same label if the adaptation functions work properly; we relax the constraint so that the conditional probability $p(y|x)$ of adapted examples should be similar across all source domains.

- The second idea is *batch entropy regularization* which maximizes the label entropy of a shuffled mini-batch. The labels of randomly selected target examples are uniformly distributed; note that we assume the target dataset is sampled from uniform label distribution. Thus, we maximize the batch entropy to prevent mode collapse where most of the target examples are mapped to a specific label.

- The third ideas are *instance entropy regularization* and *pseudo label* which minimize the label entropy of each instance. A target example naturally has a clear single label. Thus, the adapted examples should all have clear labels if the adaptation functions work properly; we minimize the label entropy after adaptation. We further bolster the entropy minimization by labeling highly confident target data with pseudo labels and minimizing cross-entropy loss between predictions and the pseudo labels.

- The last idea is *reconstruction regularization* that forces an autoencoder to reconstruct target data from the shared manifold. The autoencoder helps find the manifold without losing meaningful information. Thus, we introduce the autoencoder in DEMS with shared parameters and reconstruct target examples to learn their manifold effectively.

The overall architecture of DEMS is depicted in Fig 3. DEMS adapts the target features $X_{\mathcal{T}}$ to the source domains $\{\mathcal{S}_k\}_{k=1}^N$ via an encoder and decoders to exploit the source classifiers

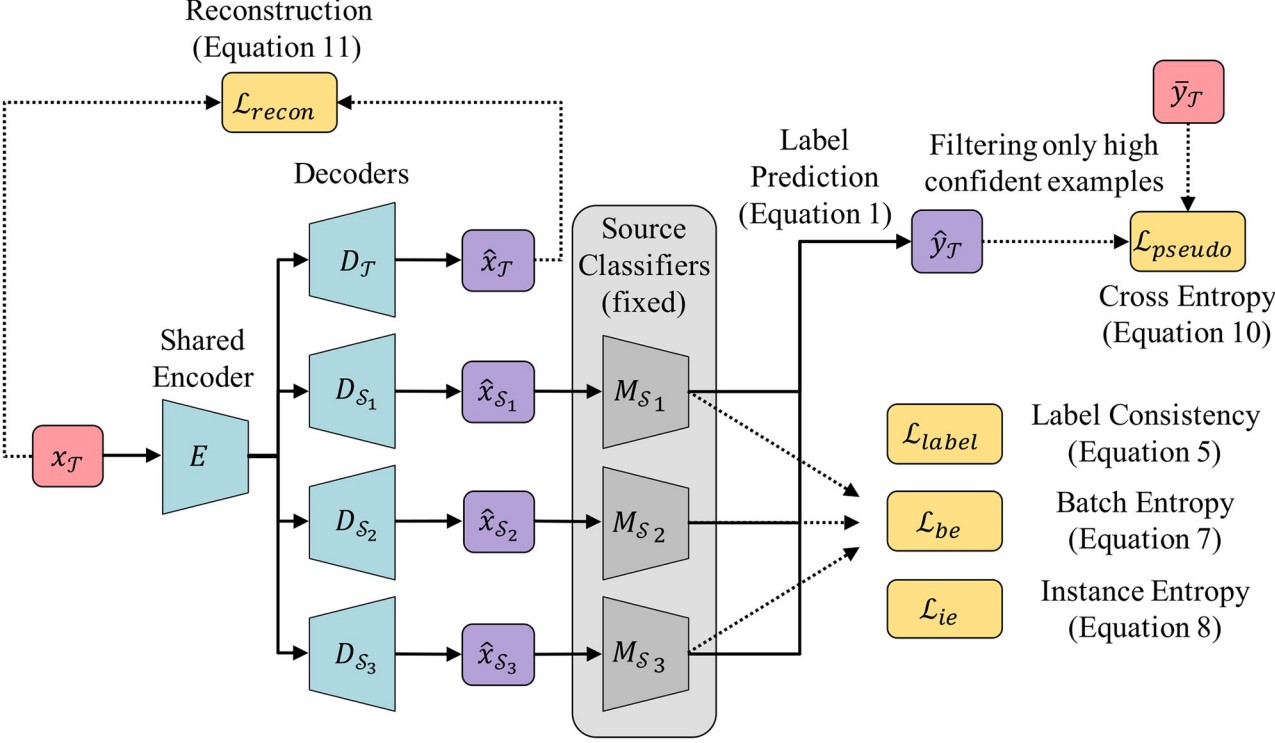

**Fig 3. Overall architecture of DEMS.**

$\{M_{\mathcal{S}_k}\}_{k=1}^{N}$. Each adaptation function $A_k : X_{\mathcal{T}} \rightarrow X_{\mathcal{S}_k}$ is divided into two components: encoder $E$ and decoder $D_{\mathcal{S}_k}$. The encoder $E$ takes a target data $x_{\mathcal{T}}$ as an input and returns its low-dimensional representation vector $z$; $E$ is shared over all domain adaptation functions. The decoder $D_{\mathcal{S}_k}$ takes the vector $z$ as an input and returns $\hat{x}_{\mathcal{S}_k}$, the translated data into the domain $\mathcal{S}_k$. Additionally, we introduce a decoder $D_{\mathcal{T}}$ that decodes the low-dimensional representation $z$ into the target domain $\mathcal{T}$. We describe the label prediction and the objective function of DEMS in the next.

## Method details

**Label prediction.** For each unlabeled target instance $x_{\mathcal{T}}$, DEMS exploits pre-trained source models $\{M_{\mathcal{S}_k}\}_{k=1}^{N}$ in predicting its label $y_{\mathcal{T}}$. Specifically, the predicted label by DEMS is formulated as:

$$\hat{y}_{\mathcal{T}} = \sum_{k=1}^{N} w_{\mathcal{S}_k} M_{\mathcal{S}_k}(\hat{x}_{\mathcal{S}_k}). \tag{1}$$

In the equation, $\hat{x}_{\mathcal{S}_k}$ is $D_{\mathcal{S}_k}(E(x_{\mathcal{T}}))$ which indicates the translated data instance into source domain $\mathcal{S}_k$ utilizing the encoder $E$ and the decoder $D_{\mathcal{S}_k}$; $0 \leq w_{\mathcal{S}_k} \leq 1$ (Eq 2) denotes the weight for the source domain $\mathcal{S}_k$. All weights add up to 1, *i.e.* $\sum_{k=1}^{N} w_{\mathcal{S}_k} = 1$, which states that DEMS predicts label $\hat{y}_{\mathcal{T}}$ of data $x_{\mathcal{T}}$ as a weighted sum of the source classifiers' predictions after domain adaptations. DEMS depends more on the prediction of a source classifier with a higher

proximity as:

$$w_{\mathcal{S}_k} = \frac{exp(\Phi(\mathcal{T}, \mathcal{S}_k)/\lambda_1)}{\sum_{k'=1}^{N} exp(\Phi(\mathcal{T}, \mathcal{S}_{k'})/\lambda_1)}, \tag{2}$$

where $\Phi(A, B)$ (Eq 3) denotes the degree of proximity between domains $A$ and $B$, and $\lambda_1 > 0$ is a hyperparameter that controls the balance of dependency on source domains. For instance, all the source classifiers contribute almost equally to the label prediction if $\lambda_1$ is a large value, while a source classifier with higher proximity $\Phi$ becomes dominant to the label prediction if $\lambda_1$ is close to 0.

It is challenging to estimate the degree of proximity between domains since data distributions $p(x)$ of domains are not observable except for the target domain. Our approach is to learn it using an objective function; the degree of proximity $\Phi(A, B)$ between domain A and B is defined by

$$\Phi(A, B) = \mathbf{v}_A^\mathsf{T} \mathbf{v}_B, \tag{3}$$

where $\mathbf{v}_A, \mathbf{v}_B \in \mathbb{R}^d$ are learnable parameters with dimensionality $d$, which indicates that the degree of proximity between domains $A$ and $B$ is estimated by an inner-product of their trained embedding vectors. The embedding vectors are trained in the optimization process.

**Objective function.** DEMS is trained to minimize the following loss:

$$\mathcal{L}_{total} = \alpha \mathcal{L}_{label} + \beta \mathcal{L}_{entropy} + \gamma \mathcal{L}_{pseudo} + \mathcal{L}_{recon}, \tag{4}$$

which consists of four different loss terms $\mathcal{L}_{label}$, $\mathcal{L}_{entropy}$, $\mathcal{L}_{pseudo}$, and $\mathcal{L}_{recon}$. $\alpha$, $\beta$, and $\gamma$ are non-negative hyperparameters that adjust the balance between the loss terms. We define these loss terms in Eqs 5, 9–11, respectively.

**Label consistency regularization.** The aim of domain adaptation is to translate domain-specific features of an example from the target domain to any source domain while preserving its semantics. If a target example $x_\mathcal{T}$ is adapted to multiple source domains while preserving its semantics, the conditional probability $p(y|x)$ of the adapted examples in all source domains should be similar. For instance, if an example has a high probability of label 4 in the target domain, the adapted example should likewise have high probabilities of label 4 in any source domain. To guarantee this property, we propose a label-consistency regularization for multi-source domain adaptation as:

$$\mathcal{L}_{label} = \binom{N}{2}^{-1} \sum_{1 \le i < j \le N} r_{\mathcal{S}_i, \mathcal{S}_j} JSD(\hat{y}_{\mathcal{S}_i} || \hat{y}_{\mathcal{S}_j}), \tag{5}$$

where $\hat{y}_{\mathcal{S}_k}$ is $M_{\mathcal{S}_k}(\hat{x}_{\mathcal{S}_k})$ indicating the label probability distribution of $x_\mathcal{T}$ estimated by source domain classifier $M_{\mathcal{S}_k}$ after adapted to the source domain $\mathcal{S}_k$. $JSD(\cdot)$ in the equation indicates Jensen-Shannon divergence [21] which is a symmetrized and smoothed version of the Kullback-Leibler divergence [22]. Jensen-Shannon divergence measures the distance between two probability distributions; a small JSD indicates that the two distributions are similar, and a large JSD indicates otherwise. $r_{\mathcal{S}_i, \mathcal{S}_j}$ (Eq 6) is a degree of proximity between $\mathcal{S}_i$ and $\mathcal{S}_j$ over the sum of all possible proximities between source domains:

$$r_{\mathcal{S}_i, \mathcal{S}_j} = \frac{exp(\Phi(\mathcal{S}_i, \mathcal{S}_j)/\lambda_2)}{\sum_{1 \le i' < j' \le N} exp(\Phi(\mathcal{S}_{i'}', \mathcal{S}_{j'})/\lambda_2)}. \tag{6}$$

$r_{\mathcal{S}_i, \mathcal{S}_j}$ strengthens label-consistency between close source domains while mitigating that

between distant source domains. $\lambda_2 > 0$ is a hyperparameter to control the degree of the regularization.

**Entropy regularizations.**   Entropy regularizations include two distinct losses based on information entropy [23]: 1) batch-entropy loss $\mathcal{L}_{be}$ for maximizing the label entropy of a batch, and 2) instance-entropy loss $\mathcal{L}_{ie}$ for minimizing the label entropy of each instance.

We assume that the target dataset is balanced against classes, *i.e.* examples are sampled with a similar probability from each label, which is a common prior for real-world data. By the assumption, the average of all target label probabilities follows a uniform distribution, *i.e.* $\left(\frac{1}{|\mathcal{C}|}, \frac{1}{|\mathcal{C}|}, \ldots, \frac{1}{|\mathcal{C}|}\right)$ where $\mathcal{C}$ denotes the set of classes. Using the fact that a uniform distribution has the maximum value of information entropy, we define the batch-entropy loss as follows:

$$\mathcal{L}_{be} = -\frac{1}{N}\sum_{k=1}^{N}H\left(\frac{1}{|\mathcal{B}|}\sum_{i\in\mathcal{B}}\hat{y}_{\mathcal{S}_k}^i\right),\tag{7}$$

where $\mathcal{B}$ is set of instances of a mini-batch $\{x_{\mathcal{T}}^i \sim p_{\mathcal{T}(x)}\}$, and $H(\cdot)$ indicates the information entropy [23];the mini-batch is also balanced against classes since it is randomly sampled from the whole dataset. By minimizing the batch-entropy loss, we force the average of batch-wise label probabilities estimated by each source classifier after adaptation to have a uniform probability distribution.

On another aspect, each target instance inherently has a clear single label, which indicates that it has a one-hot label probability even if the exact label probability is unknown. Based on the fact that a one-hot probability distribution has the minimum value of information entropy [23], we define the instance-entropy loss as follows:

$$\mathcal{L}_{ie} = \frac{1}{N|\mathcal{B}|}\sum_{k=1}^{N}\sum_{i\in\mathcal{B}}H(\hat{y}_{\mathcal{S}_k}^i).\tag{8}$$

We finally define the total entropy loss by summing up batch-entropy loss (Eq 7) and instance-entropy loss (Eq 8) as follows:

$$\mathcal{L}_{entropy} = \mathcal{L}_{be} + \mathcal{L}_{ie}.\tag{9}$$

**Pseudo label.**   High confidence of the predicted label of a target example, which is estimated by Eq 1, indicates that the example is successfully adapted to source domains and clearly classified by the source classifiers. Accordingly, we employ pseudo-labels to bolster the current predictions by pretending that the predicted label is the ground-truth label. The pseudo-label loss is formulated by a cross-entropy between the predictions and the pseudo-labels as follows:

$$\mathcal{L}_{pseudo} = -\frac{1}{|b|}\sum_{i\in b}\sum_{j\in\mathcal{C}}(\bar{y}_{\mathcal{T}}^i)_j\log{(\hat{y}_{\mathcal{T}}^i)_j},\tag{10}$$

where $\mathcal{C}$ is the set of classes, $\bar{y}_{\mathcal{T}} = Dirac(\hat{y}_{\mathcal{T}})$, and $(y)_j$ denotes the probability of $j$-th class in $y$. $\hat{y}_{\mathcal{T}}$ is a predicted target label by DEMS (Eq 1). $Dirac(\cdot)$ is a function that makes a Dirac distribution; for simplicity, we choose one-hot vectorization that sets the maximum probability to 1 and the rest to 0. Only examples that meet $\max_j(\hat{y}_{\mathcal{T}})_j > \epsilon$, where $0 \leq \epsilon \leq 1$ is a hyperparameter that regulates the threshold of confidence, are sampled from the mini-batch $\mathcal{B}$; $b \subset \mathcal{B}$ in Eq 10 indicates the selected subset of the mini-batch.

**Reconstruction.**   Autoencoders [24], which encode input data to low-dimensional vectors and decode them into the original space by reconstruction regularization, learn a meaningful

low-dimensional manifold by preventing the simple copy of the input data. We employ an autoencoder sharing the encoder $E$ in finding a low-dimensional manifold $z$. The reconstruction loss is formulated as follows:

$$\mathcal{L}_{recon} = |x_T - \hat{x}_T|_1, \tag{11}$$

where $\hat{x}_T$ is $D_T(E(x_T))$ indicating the reconstruction of $x_T$ by encoder $E$ and decoder $D_T$, and $\|\cdot\|_1$ denotes the $l_1$ norm.

**Algorithm 1** Training DEMS (Data-free Exploitation of Multiple Sources)

**Require:** unlabeled target dataset $X_T = \{x_T^i\}_{i=1}^{n_T}$
**Require:** trained source classifiers $\{M_{S_k} : x_{S_k} \to y_{S_k}\}_{k=1}^{N}$
**Require:** adaptation networks $\{A_k : X_T \to X_{S_k}\}_{k=1}^{N}$
**Require:** hyperparameters $\alpha$, $\beta$, $\gamma$, $\lambda_1$, $\lambda_2$, and $\epsilon$
**Ensure:** trained adaptation networks $\{A_k : X_T \to X_{S_k}\}_{k=1}^{N}$
1: **for** [1, num_epochs] **do**
2:   Calculate the label consistency loss $\mathcal{L}_{label}$ (Eq 5)
3:   Calculate the batch-entropy loss $\mathcal{L}_{be}$ (Eq 7)
4:   Calculate the instance-entropy loss $\mathcal{L}_{ie}$ (Eq 8)
5:   Calculate the entropy loss $\mathcal{L}_{entropy} \leftarrow \mathcal{L}_{be} + \mathcal{L}_{ie}$ (Eq 9)
6:   Predict the target labels $\hat{y}_T$ (Eq 10) and filter only ones that meet $\max_j(\hat{y}_T)_j > \epsilon$
7:   Calculate the pseudo-label loss $\mathcal{L}_{pseudo}$ (Eq 10)
8:   Calculate the reconstruction loss $\mathcal{L}_{recon}$ (Eq 11)
9:   Calculate the total loss $\mathcal{L}_{total} \leftarrow \alpha\mathcal{L}_{label} + \beta\mathcal{L}_{entropy} + \gamma\mathcal{L}_{pseudo} + \mathcal{L}_{recon}$ (Eq 4)
10:   Update the parameters of $\{A_k\}_{k=1}^{N}$ to minimize $\mathcal{L}_{total}$
11: **end for**

**Algorithm.** We summarize the training algorithm of DEMS in Algorithm 1. DEMS takes initialized adaptation networks $\{M_{S_k} : x_{S_k} \to y_{S_k}\}_{k=1}^{N}$ and trains them while exploiting pre-trained source classifiers without any source data. DEMS calculates the total loss $\mathcal{L}_{total}$ in lines 2 to 9. Then, in line 10, DEMS updates the parameters of the adaptation networks $\{M_{S_k}\}_{k=1}^{N}$ to minimize the total loss $\mathcal{L}_{total}$. This is repeated until the adaptation networks $\{M_{S_k}\}_{k=1}^{N}$ are trained properly; we use validation set and the training is performed until the total loss $\mathcal{L}_{total}$ of the validation set is the lowest. After being trained, DEMS predicts the target labels of test data by Eq 10 using the trained adaptation networks. The predicted target labels are evaluated by the ground-truth labels and we report the accuracies in the next section. The computational complexity is dependent on the architecture of the encoder and decoders. In the case of a CNN-based architecture, the computational complexity of label prediction of DEMS is $\mathcal{O}(HWk^2MN)$; $H$ and $W$ are height and width of input image, respectively, $k$ is size of kernel, and $M$ and $N$ are sizes of input and output channels, respectively.

## Experiments

We conduct experiments to answer the following questions:

- **Q1. Accuracy.** How accurate is DEMS on real-world datasets?

- **Q2. Qualitative analysis.** How well does DEMS adapt a given target example to source domains?

- **Q3. Parameter sensitivity.** How much do $\epsilon$ (Eq 10) and $\lambda$ (Eqs 2 and 6) affect the accuracy?

 

**Table 4. Summary of datasets.**

| Dataset | Features | Training | Validation | Test |
|---|---|---|---|---|
| MNIST | $1 \times 28 \times 28$ | 55,000 | 5,000 | 10,000 |
| MNIST-M | $3 \times 32 \times 32$ | 55,000 | 5,000 | 10,000 |
| SVHN | $3 \times 32 \times 32$ | 68,257 | 5,000 | 26,032 |
| SynDigits | $3 \times 32 \times 32$ | 55,000 | 5,000 | 9,553 |
| USPS | $1 \times 16 \times 16$ | 6,291 | 1,000 | 2,007 |

## Experimental settings

**Datasets.** We use five different number datasets: MNIST [25], MNIST-M [10], SVHN [26], SynDigits [27], and USPS [28], which are summarized in Table 4; Fig 4 shows sample images of each dataset. For SynDigits, we use a randomly selected subset of 60,000 images for training and validation out of 479,400 images;the subset is considered to possess sufficient domain knowledge since a classifier trained on it shows 95.9% accuracy. We use the original datasets for the other datasets. The five datasets are scaled to the size of ($3 \times 32 \times 32$) to have the same input dimensionality. We set one of them as a target and the rest as sources in the experiments.

**Baselines.** We set three baselines: *Best single source*, *Average*, and *Weighted sum*. *Best single source* directly feeds the target data into source classifiers, and the source classifier which yields the best performance is chosen. *Average* feeds the target data into all source classifiers and averages the resulting label probabilities to predict target labels. *Weighted sum* takes a weighted sum of the results after feeding the target data into source classifiers; we utilize Eq 2 for the weights, and set $\Phi(\mathcal{T}, \mathcal{S}_k)$ as $\xi - \mathcal{L}_{entropy}^{\mathcal{T} \rightarrow \mathcal{S}_k}$, where $\mathcal{L}_{entropy}^{\mathcal{T} \rightarrow \mathcal{S}_k}$ is the sum of batch-entropy loss and instance-entropy loss that are estimated when the target data are directly fed into source classifier $M_{\mathcal{S}_k}$. $\xi$ is a hyperparmeter and we set it to 1 for all experiments. The intuition behind the definition of $\Phi(\mathcal{T}, \mathcal{S}_k)$ is that $\mathcal{L}_{entropy}^{\mathcal{T} \rightarrow \mathcal{S}_k}$ is presumable to be low if the degree of proximity between $\mathcal{T}$ and $\mathcal{S}_k$ is high.

**Network architecture.** We pre-train ResNet14 [29] for each dataset to generate the source classifiers. We adopt the architecture of generator in CycleGAN [30]; the encoder is composed of two convolutional layers with stride size two and three residual blocks [29]; each of the decoder is composed of three residual blocks and two transposed convolutional layers with stride size two. We use batch normalization [31] for the encoder and the decoders. Note that an appropriate network architecture should be selected for each domain of application;

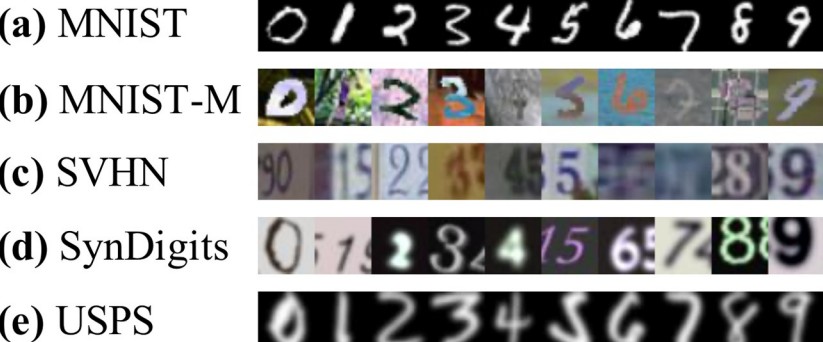

**(a)** MNIST

**(b)** MNIST-M

**(c)** SVHN

**(d)** SynDigits

**(e)** USPS

**Fig 4. Sample images (10 classes).**

**Table 5. Classification accuracy of DEMS and baselines.**

| Target dataset | Best single source (Single source) | Average (Multi-sources) | Weighted sum (Multi-sources) | DEMS (proposed) (Multi-sources) |
|---|---|---|---|---|
| MNIST | 97.65 ± 0.75% | 94.87 ± 1.22% | 96.37 ± 0.40% | **99.01 ± 0.12%** |
| MNIST-M | 45.03 ± 3.74% | 33.50 ± 1.72% | 40.91 ± 1.24% | **72.57 ± 3.20%** |
| SVHN | 71.87 ± 3.53% | 23.11 ± 1.61% | 56.09 ± 5.17% | **76.60 ± 1.39%** |
| SynDigits | 91.89 ± 1.79% | 60.47 ± 5.69% | 78.66 ± 4.37% | **93.74 ± 0.79%** |
| USPS | 82.03 ± 3.77% | 84.54 ± 5.31% | 88.09 ± 2.04% | **96.14 ± 0.41%** |

The remains except for the target dataset are used for sources. The best method is in bold, and the second best one is underlined. Note that DEMS gives the best performance.

recurrent neural networks [32] and graph autoencoders [33] could be selected in the natural language processing domain [34, 35] and in the graph domain [36–39], respectively.

**Training details.** We first minimize $\mathcal{L}_{recon}$ during the first 5 epochs, initialize $\{D_{\mathcal{S}_k}\}_{k=1}^{N}$ with the trained $D_{\mathcal{T}}$, and then minimize $\mathcal{L}_{total}$. Finally, a classification accuracy of the test target dataset is reported at the lowest validation loss $\mathcal{L}_{total}$ among 100 epochs. Each experiment is performed 5 times with different random seeds, and the standard deviation is reported along with the average. We use the hyperparameters that give the best performance. We set $\alpha = 0.1$, $\beta = 1$, and $\gamma = 1$ among {0.1, 0.5, 1, 5, 10} in Eq 4. Unless otherwise noted, $\epsilon$ (Eq 10) is set to 0.9 among {0.1, 0.2, 0.3, 0.4, 0.5, 0.6, 0.7, 0.8, 0.9}. We set $\lambda_1$ (Eq 2) and $\lambda_2$ (Eq 6) the same as $\lambda$; $\lambda$ is set to 1 among {0.125, 0.25, 0.5, 1, 2, 4, 8}. We set the dimensionality of $\mathbf{v}_A$ and $\mathbf{v}_B$ as 10 in Eq 3. All the networks are trained with Adam optimizer [40] with learning rate 0.001, $l_2$ regularization coefficient 0.0001, $\beta_1 = 0.9$, and $\beta_2 = 0.999$. We implement all the codes with PyTorch and perform a grid search to find the best hyperparameters, using a workstation with RTX 2080 Ti.

## Accuracy

**Overall performance.** We compare DEMS with other baselines for data-free UMDA. Table 5 shows the classification accuracy. DEMS shows the best performance outperforming the baselines in all experiments. In particular, the performance differences between DEMS and the baselines are large for the MNIST-M target which has very complex patterns as shown in Fig 4; DEMS shows 27.5% point higher accuracy than the best baseline. In all experiments except the USPS target, *Average* and *Weighted sum* exploiting the knowledge of multiple source domains show worse performances than *Best single source* exploiting the knowledge of single source domain. This demonstrates how challenging data-free UMDA problem is and supports the contribution of this work.

**Ablation study.** We conduct an ablation study to evaluate how each loss of DEMS contributes to the performance. Table 6 shows the ablation study that evaluates the effectiveness of each loss in DEMS. Note that each of the proposed losses in the objective function (Eq 4) contributes significantly to the performance of DEMS, showing the effectiveness of our ideas.

**Table 6. Ablation study on MNIST-M target dataset.**

| DEMS | DEMS − $\mathcal{L}_{label}$ | DEMS − $\mathcal{L}_{be}$ | DEMS − $\mathcal{L}_{ie}$ | DEMS − $\mathcal{L}_{pseudo}$ | DEMS − $\mathcal{L}_{recon}$ |
|---|---|---|---|---|---|
| **72.69 ± 2.60%** | 65.65 ± 5.55% | 10.33 ± 0.62% | 59.58 ± 3.57% | 43.88 ± 1.73% | 11.11 ± 1.59% |

DEMS − $\mathcal{L}$ indicates a variant of DEMS with $\mathcal{L}$ excluded from $\mathcal{L}_{total}$. Note that each of the loss significantly contributes to the accuracy of DEMS.

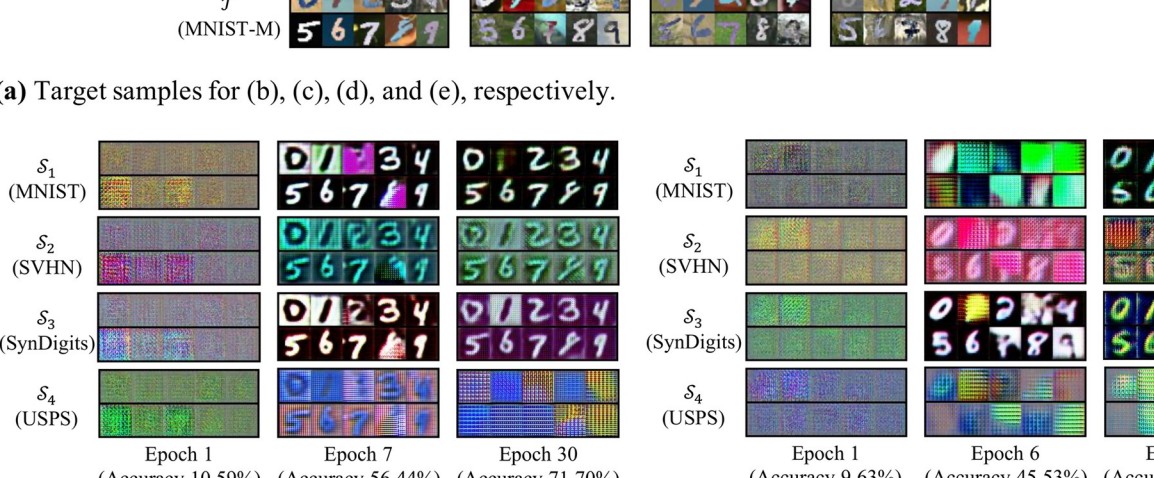

**(a)** Target samples for (b), (c), (d), and (e), respectively.

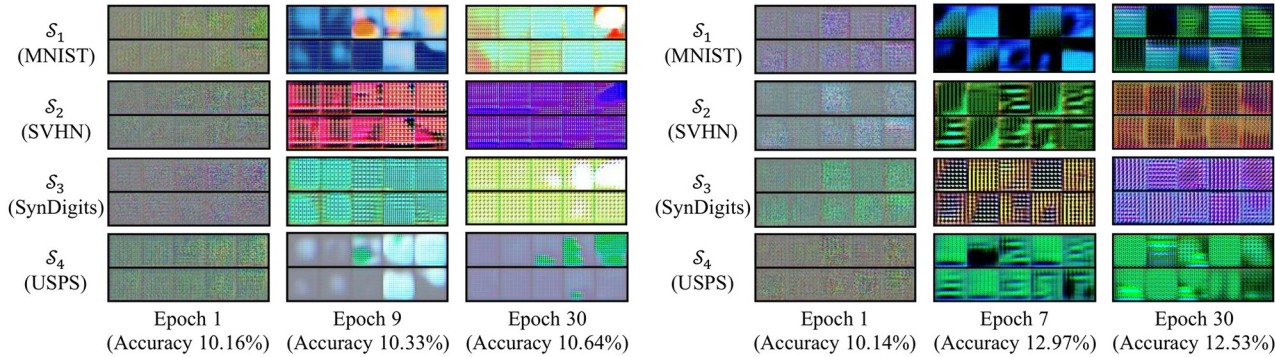

**(b)** DEMS which includes the whole regularizations.

**(c)** DEMS-$\mathcal{L}_{pseudo}$ which excludes $\mathcal{L}_{pseudo}$ from $\mathcal{L}_{total}$.

**(d)** DEMS-$\mathcal{L}_{be}$ which excludes $\mathcal{L}_{be}$ from $\mathcal{L}_{total}$.

**(e)** DEMS-$\mathcal{L}_{recon}$ which excludes $\mathcal{L}_{recon}$ from $\mathcal{L}_{total}$.

**Fig 5. Visualization of image adaptation from MNIST-M to other source domains.** Fig (a) enumerates target samples for Figs (b), (c), (d), and (e). The target samples are adapted by adaptation networks which are trained with different losses. For DEMS (Fig (b)), the adaptation gradually focuses on the close source domains (MNIST, SVHN, and SynDigits), resulting in performance enhancement. For DEMS $- \mathcal{L}_{pseudo}$ (Fig (c)), some classes (digits 3, 7, and 9) are failed to be adapted to source domains. For DEMS $- \mathcal{L}_{be}$ and DEMS $- \mathcal{L}_{recon}$ (Figs (d) and (e)), the adaptations are not trained at all.

## Qualitative analysis

We analyze DEMS and its variants DEMS $- \mathcal{L}$ qualitatively to evaluate how well DEMS adapts data to different domains; DEMS $- \mathcal{L}$ indicates a variant of DEMS with $\mathcal{L}$ excluded from $\mathcal{L}_{total}$. Note that the baseline algorithms are not analyzed qualitatively since they do not adapt data to different domains (see Table 1). For DEMS $- \mathcal{L}$, we select three variants DEMS $- \mathcal{L}_{pseudo}$, DEMS $- \mathcal{L}_{be}$, and DEMS $- \mathcal{L}_{recon}$ which show the lowest accuracies in the ablation study (see Table 6).

Fig 5 visualizes adapted sample examples from MNIST-M to MNIST, SVHN, SynDigits, and USPS, respectively. DEMS (Fig 5b) translates the images into noises at the beginning of training (epoch 1). As training progresses, however, meaningful patterns (*e.g.* shape of digits rather than backgrounds) of the target images are detected and adapted to each source domain (epoch 7). As training progresses more (epoch 30), DEMS focuses adaptation on closer source domains (MNIST, SVHN, and SynDigits) than to the far source domain (USPS), and its

classification performance improves. DEMS $- \mathcal{L}_{pseudo}$ (Fig 5c) successfully adapts most of the classes to MNIST and SynDigits, but fails to adapt some classes (digits 3, 7, and 9) to the source domains yielding degraded classification performance. It is shown that DEMS $- \mathcal{L}_{be}$ (Fig 5d) and DEMS $- \mathcal{L}_{recon}$ (Fig 5e) do not learn to adapt the target data to the source domains.

## Parameter sensitivity

**Sensitivity of $\epsilon$.** The hyperparameter $\epsilon$, which is involved in $\mathcal{L}_{pseudo}$ (Eq 10), governs the threshold of pseudo-labels. As $\epsilon$ increases, the selected examples have higher confidence while fewer examples are selected. On the other hand, as $\epsilon$ decreases, the number of selected examples increases while the confidence of the examples decreases. As shown in Fig 6a, the accuracy is the highest when $\epsilon$ is 0.9 for all datasets, and the accuracy is significantly reduced in the extreme case when $\epsilon = 1$. The results demonstrate that DEMS is best optimized through high-quality pseudo-labels.

**Sensitivity of $\lambda$.** The hyperparameter $\lambda$, which is involved in Eqs 2 and 6, controls the balance of dependency between domains; note that $\lambda_1 = \lambda_2 = \lambda$ for our experiments. For instance, if $\lambda$ is a large positive value, all the source classifiers almost equally contribute to the target label prediction in Eq 1 and are highly regulated to output the similar predictions in Eq 5. For instance, if $\lambda$ is a large positive value, all the source classifiers almost equally contribute to the target label prediction in Eq 1 and even source classifiers that are not close to each other are regulated to output the similar predictions in Eq 5. Conversely, if $\lambda$ is close to zero, a source classifier closer to the target domain contributes more to the target label prediction in Eq 1 and source classifiers that are not closer to each other are less regulated to output similar predictions in Eq 5. Fig 6b shows that the best results are obtained when $\lambda = 1$ for all target domains, and the performance degrades if the $\lambda$ is too large or too small. In particular, SVHN which has relatively complex patterns shows a severely degraded performance when $\lambda$ is larger

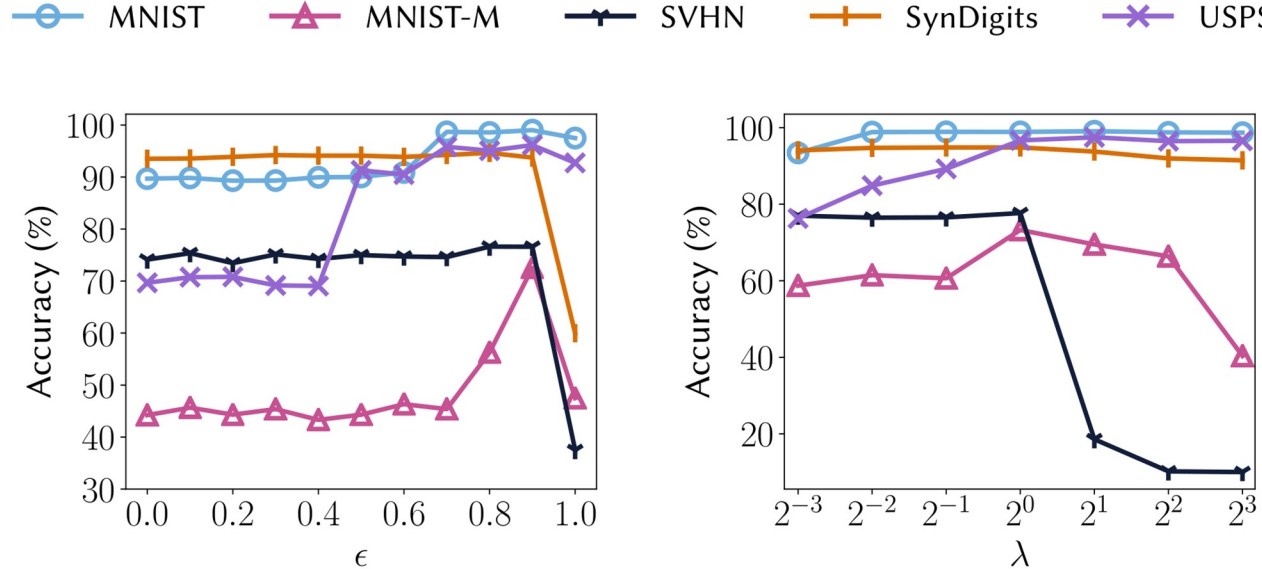

**(a)** Sensitivity to threshold $\epsilon$.　　**(b)** Sensitivity to temperature $\lambda$.

**Fig 6. Sensitivity of accuracy to the hyperparameters $\epsilon$ (Eq 10) and $\lambda$ (Eqs 2 and 6).**

than 2, which means that it is more helpful for a complex target to consider a nearby source than all sources.

## Conclusion

We propose DEMS (Data-free Exploitation of Multiple Sources), a novel architecture for multi-source domain adaptation without any observable source data. DEMS learns to adapt target data to each source domain to exploit the pre-trained source classifiers. Experiments on real-world datasets show that DEMS outperforms baselines up to 27.5% point higher accuracy, by successfully learning the adaptation function and exploiting the source classifiers in target label predictions. However, DEMS assumes that the source and target domains have similar feature spaces and have the same label space. Thus, DEMS is not applicable in domain adaptation between heterogeneous domains. Future works include extending DEMS to transfer knowledge between heterogeneous domains, *e.g.* from images to text or vice versa, that may require careful design of adaptation networks.

## Author Contributions

**Conceptualization:** Hyunsik Jeon.

**Formal analysis:** Hyunsik Jeon.

**Investigation:** Seongmin Lee.

**Methodology:** Hyunsik Jeon, Seongmin Lee.

**Software:** Hyunsik Jeon.

**Validation:** Hyunsik Jeon.

**Visualization:** Hyunsik Jeon.

**Writing – original draft:** Hyunsik Jeon.

**Writing – review & editing:** Hyunsik Jeon, Seongmin Lee, U Kang.

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
