## [Decision Letter · Decision Letter 0]

26 Apr 2021

PONE-D-21-11372

Unsupervised Multi-Source Domain Adaptation with No Observable Source Data

PLOS ONE

Dear Dr. Kang,

Thank you for submitting your manuscript to PLOS ONE. After careful consideration, we feel that it has merit but does not fully meet PLOS ONE’s publication criteria as it currently stands. Therefore, we invite you to submit a revised version of the manuscript that addresses the points raised during the review process.

Based on the comments received from the reviewers and my own observation, I recommend minor revisions for the paper.

We look forward to receiving your revised manuscript.

Kind regards,

Thippa Reddy Gadekallu

Academic Editor

PLOS ONE

Journal Requirements:

Reviewers' comments:

Reviewer's Responses to Questions

**Comments to the Author**

1. Is the manuscript technically sound, and do the data support the conclusions?

Reviewer #1: Yes

Reviewer #2: Yes

2. Has the statistical analysis been performed appropriately and rigorously? 

Reviewer #1: Yes

Reviewer #2: Yes

3. Have the authors made all data underlying the findings in their manuscript fully available?

Reviewer #1: Yes

Reviewer #2: Yes

4. Is the manuscript presented in an intelligible fashion and written in standard English?

Reviewer #1: Yes

Reviewer #2: Yes

5. Review Comments to the Author

Reviewer #1: - Please highlight the contribution clearly in the introduction

- this paper lacks in Novelty of the proposed approach. The author should highlight the contribution clearly in the introduction and provide a comparison note with existing studies.

- Some Paragraphs in the paper can be merged and some long paragraphs can be split into two.

- The quality of the figures can be improved more. Figures should be eye-catching. It will enhance the interest of the reader.

- What are the computational resources reported in the state of the art for the same purpose?

- Please cite each equation and clearly explain its terms.

- Math work should be written math mode.

- What are the evaluations used for the verification of results?

- Clearly highlight the terms used in the algorithm and explain them in the text.

- Authors should add the most recent reference:

1) DARE-SEP: A Hybrid Approach of Distance Aware Residual Energy-Efficient SEP for WSN, IEEE Transactions on Green Communications and Networking

2) Interference Mitigation in D2D Communication Underlying Cellular Networks: Towards Green Energy, Computers, Materials & Continua

Reviewer #2: 1. What are the limitations of the existing works that motivated the current research?

2. Summarize the key findings from the related works in the form of a table.

3. Some of the recent works such as teh following on DNN/ML can be discussed in the paper: "An Eﬀective Feature Engineering for DNN using Hybrid PCA-GWO for Intrusion Detection in IoMT Architecture, An optimal transportation routing approach using GIS-based dynamic traffic flows"

3. Present computational complexity of the proposed approach.

4.Compare the current work with recent state-of-the-art.

5. Discuss about the limitations of the current work in conclusion.

6. PLOS authors have the option to publish the peer review history of their article (what does this mean?). If published, this will include your full peer review and any attached files.

Reviewer #1: No

Reviewer #2: No

---

## [Author Response · Author response to Decision Letter 0]

3 Jun 2021

1. Reviewer 1.

• (R1-1) Please highlight the contribution clearly in the introduction.

– (A1-1) The contents of the contribution list have been revised to clearly highlight the contributions (lines 38-40 and lines 43-44 in introduction section).

• (R1-2) This paper lacks in novelty of the proposed approach. The author should highlight the contribution clearly in the introduction and provide a comparison note with existing studies.

– (A1-2) We revised the contribution list (lines 38-40 and lines 43-44 in introduction section) and provided a comparison note with the competitors in the introduction (lines 29-34 in introduction section). The existing studies for unsupervised multi-source domain adaptation (UMDA) assume that source data are observable and they train the adaptation networks to align manifolds of source and target domains. Thus, the existing methods are not applicable to our setting where source data are not observable. On the other hand, DEMS trains the adaptation networks while regulating the results of the source classifiers (lines 67-72 in related work section).

• (R1-3) Some Paragraphs in the paper can be merged and some long paragraphs can be split into two.

– (A1-3) We reviewed the overall manuscript to reorganize the paragraphs in it. Especially, we itemized the main ideas which were in one long paragraph into individual items (lines 104-126 in proposed method section).

• (R1-4) The quality of the figures can be improved more. Figures should be eye-catching. It will enhance the interest of the reader.

– (A1-4) We improved Figure 3, which previously looked complicated, to be eye-catching.

• (R1-5) What are the computational resources reported in the state-of-the-art for the same purpose?

– (A1-5) We implemented all the codes using PyTorch and trained all networks including DEMS and competitors using RTX 2080 Ti (lines 288-289 in experiments section).

• (R1-6) Please cite each equation and clearly explain its terms.

– (A1-6) We cited all equations in the manuscript and clearly explained the terms (lines 143, 147, 164, 179, 203, 204, and 213 in experiments section).

• (R1-7) Math work should be written math mode.

– (A1-7) We already wrote the math works by math mode. If there is anything we missed, please let us know and we will fix it.

• (R1-8) What are the evaluations used for the verification of results?

– (A1-8) We used validation set and trained DEMS until the loss L_{total} of the validation set is the lowest. Each experiment is performed 5 times with different random seeds and we reported the standard deviation along with the averaged accuracy. We added such description in lines 232-233 in proposed method section and lines 278-281 in experiments section.

• (R1-9) Clearly highlight the terms used in the algorithm and explain them in the text.

– (A1-9) We summarized the training algorithm of DEMS and explained the process in algorithm section (lines 226-240 in proposed method section).

• (R1-10) Authors should add the most recent reference: 1) DARE-SEP: A Hybrid Approach of Distance Aware Residual Energy-Efficient SEP for WSN, IEEE Transactions on Green Communications and Networking, and 2) Interference Mitigation in D2D Communication Underlying Cellular Networks: Towards Green Energy, Computers, Materials & Continua.

– (A1-10) We added the two references (lines 273-276 in experiments section).

2. Reviewer 2.

• (R2-1) What are the limitations of the existing works that motivated the current research?

– (A2-1)Previous works have focused on unsupervised multi-source domain adaptation(UMDA) where source data are accessible. Thus, they trained the adaptation networks to align manifolds between source and target domains using the source and the target data. However, source data are not easily accessible in practical scenarios although source classifiers are readily accessible. Hence, we are motivated to develop a method to train adaptation networks using the source classifiers and the target data without using the source data. We added such discussion in lines 67-72 in related works section.

• (R2-2) Summarize the key findings from the related works in the form of a table.

– (A2-2) We summarized the key findings from the related works and compared them with our proposed method in Table 3 (lines 61-62 in related works section).

• (R2-3) Some of the recent works such as the following on DNN/ML can be discussed in the paper: ”An Effective Feature Engineering for DNN using Hybrid PCA-GWO for Intrusion Detection in IoMT Architecture”, and ”An optimal transportation routing approach using GIS-based dynamic traffic flows”

– (A2-3) We added the two references (lines 273-276 in experiments section).

• (R2-4) Present computational complexity of the proposed approach.

– (A2-4) We added the computational complexity of DEMS (lines 236-240 in proposed method section).

• (R2-5) Compare the current work with recent state-of-the-art.

– (A2-5) The data-free UMDA is a novel problem since there were no previous approaches which can work without the source data. Nevertheless, we introduced several baselines in Table 1 and explained them in the introduction (lines 29-34 in introduction section).

• (R2-6) Discuss about the limitations of the current work in conclusion.

– (A2-6) We included the limitations of the work in conclusion section (lines 351-354 in conclusion section).

---

## [Decision Letter · Decision Letter 1]

7 Jun 2021

Unsupervised Multi-Source Domain Adaptation with No Observable Source Data

PONE-D-21-11372R1

Dear Dr. Kang,

We’re pleased to inform you that your manuscript has been judged scientifically suitable for publication and will be formally accepted for publication once it meets all outstanding technical requirements.

Kind regards,

Thippa Reddy Gadekallu

Academic Editor

PLOS ONE

Additional Editor Comments (optional):

Reviewers' comments:

Reviewer's Responses to Questions

**Comments to the Author**

1. If the authors have adequately addressed your comments raised in a previous round of review and you feel that this manuscript is now acceptable for publication, you may indicate that here to bypass the “Comments to the Author” section, enter your conflict of interest statement in the “Confidential to Editor” section, and submit your "Accept" recommendation.

Reviewer #1: All comments have been addressed

Reviewer #2: All comments have been addressed

2. Is the manuscript technically sound, and do the data support the conclusions?

Reviewer #1: Yes

Reviewer #2: Yes

3. Has the statistical analysis been performed appropriately and rigorously? 

Reviewer #1: Yes

Reviewer #2: Yes

4. Have the authors made all data underlying the findings in their manuscript fully available?

Reviewer #1: Yes

Reviewer #2: Yes

5. Is the manuscript presented in an intelligible fashion and written in standard English?

Reviewer #1: Yes

Reviewer #2: Yes

6. Review Comments to the Author

Reviewer #1: The authors have addressed almost all my suggestions. I would like to accept this paper.

Reviewer #2: The authors have done a good job in addressing all the comments and suggestions. The paper is improved significantly and is in a good shape now. I recommend the paper to be accepted in the current form.

7. PLOS authors have the option to publish the peer review history of their article (what does this mean?). If published, this will include your full peer review and any attached files.

Reviewer #1: No

Reviewer #2: No

---

## [Editor Report · Acceptance letter]

29 Jun 2021

PONE-D-21-11372R1 

Unsupervised Multi-Source Domain Adaptation with No Observable Source Data  

Dear Dr. Kang:

I'm pleased to inform you that your manuscript has been deemed suitable for publication in PLOS ONE. Congratulations! Your manuscript is now with our production department. 

Kind regards, 

on behalf of

Dr. Thippa Reddy Gadekallu 

Academic Editor

PLOS ONE